# Over-indebtedness and health in Switzerland: A cross-sectional study comparing over-indebted individuals and the general population

**Oliver Hämmig**[1]*, **Joanna Herzig**[2]

**1** Epidemiology, Biostatistics and Prevention Institute, University of Zurich, Zurich, Switzerland, **2** Debt. Prevention, City of Zurich, Switzerland

* oliver.haemmig@uzh.ch

## Abstract

### Background

Previous international studies have shown that over-indebtedness is associated with poor health. However, in Switzerland research addressing over-indebtedness is widely lacking, strongly needed and particularly important because it is evidently a rising but still commonly tabooed, socially "undesired" and highly stigmatized phenomenon that is rarely discussed and largely ignored and unexplored.

### Methods

A cross-sectional survey was conducted among over-indebted adults seeking advice from one of the four official debt advisory centers in the Canton of Zurich. The survey finally included 219 respondents participating voluntarily and anonymously. This sample was then linked with a comparable subsample of the nationally representative Swiss Health Survey of 2017, namely 1,997 respondents of the same age from the Canton of Zurich. For reasons of comparability identical health questions and measures were taken from the Swiss Health Survey and used in the over-indebtedness survey. The pooled or combined dataset covered a total of 2,216 adult individuals.

### Results

Remarkably high prevalence rates and relative risks of poor self-rated health, severe musculoskeletal and sleep disorders and moderate to severe depression were observed among over-indebted individuals compared to the general population. More than 50% of the over-indebted individuals had poor general health or moderate to severe depression compared to the general population with 14% and 7%, respectively. And far above one third of the over-indebted but 'only' between 6% and 8% of the general population showed severe musculo-skeletal disorders and sleep disorders. Even after adjustment for various control variables and covariates, over-indebtedness increased the odds ratios for poor health outcomes consistently and dramatically, i.e. by a factor of 8 and more (aOR = 8.5–11.6).

**Data Availability Statement:** All relevant data are within the paper and its Supporting information files.

**Funding:** The author(s) received no specific funding for this work.

**Competing interests:** The authors have declared that no competing interests exist.

## Conclusions

Over-indebtedness in Switzerland has particularly negative effects on various aspects of the health of the persons concerned, irrespective of their demographic characteristics and their social and employment status.

## Introduction

Indebtedness and particularly over-indebtedness are growing phenomena worldwide and have become a major social problem in the past decades, not only but particularly in high-income countries and affluent societies across Europe and Northern America [1–8]. Along with the rise of working poor since the early 1990s and at the latest as a consequence of the financial crisis and the subsequent economic recession and job losses in the late 2000s, a remarkable increase in the number and a growing proportion of households and individuals who are in arrears with outstanding payments and at risk of over-indebtedness has been observed and reported [9, 10].

Over-indebtedness usually results from individual behaviors and personal circumstances, namely from a persistently low household income which is insufficient for covering the regular expenditures (e.g. working poverty, single parenthood), from poor money management and excessive consumption (over-spending), and/or from critical life events (e.g. unemployment, separation, divorce, illness) [9].

Over-indebtedness is commonly understood and defined as the impossibility to repay all debts completely and on time, or more precisely, as an ongoing rather than a temporary inability to meet financial obligations and commitments and recurring expenses [9, 10]. Over-indebtedness means too much debt in relation to the household income or in other words, a situation where household income, in spite of a reduction in the living standard, is insufficient to meet all payment obligations over a longer period [11]. In contrast to indebtedness, which is normal and not problematic per se, over-indebtedness is considered to be problematic in many ways.

There are different aspects and definitions of over-indebtedness and accordingly a number of indicators or criteria that need to be applied in order to measure or to speak of over-indebtedness [12]. These aspects that characterize and indicate over-indebtedness include the following: (1) making high repayments relative to income, i.e. spending a large proportion of the household income on repayments or debt service payments, (2) being in arrears with repayments or with servicing the debts, (3) making heavy use of credits or rather having multiple debts (loans or credit commitments) and, finally, (4) finding debt and/or debt service a heavy burden financially and/or psychologically. Close to the latter is an additional indicator or aspect, namely seeking professional assistance or debt advice to better cope with debts and financial obligations.

For Switzerland as one of the wealthiest countries in Europe, over-indebtedness is not even recognized as an increasing phenomenon and a societal problem, and as a consequence there is hardly any individual and cross-sectional data and a lack of population-based and nationally representative statistics on debt and even more on over-indebtedness. However, there are indications that the number of people with debt problems is rising in Switzerland. An analysis of the online comparison service comparis.ch recently showed for 2019 an over-indebtedness rate of 6.5% in the Swiss population, or 561,000 people who were unable to service their debts. It further showed an increase of this number by 22% within only three years, i.e. since 2016.

This is quite a large number and a remarkable trend, considering that Switzerland is one of the very few countries in Europe that does not have an official debt relief procedure for over-indebted individuals.

According to the Swiss Federal Office of Statistics and the Statistics on Income and Living Conditions (SILC) of 2013, nearly 8% of the total population in Switzerland have at least *three kinds of debts* (vehicle leasing debt, consumer loans, overdrafts, instalment payments, personal loans from friends or relatives, etc.) and just over 8% have at least *two kinds of payments in arrears* (on bills, rents, loans, taxes, alimony payments, health insurance premiums, etc.). In other words, almost every twelfth person living in Switzerland has different debts and debtors and/or more than one payment in arrears and is therefore at risk of over-indebtedness, regardless of the total amount of debt and their income. Not surprisingly, these proportions are almost twice as high up to three times higher than average among people with low educational attainment, young people under the age of 25, single parent households, low-income earners, foreigners from outside Europe, or the unemployed.

But most importantly, over-indebtedness represents not only a high poverty risk but also an increased health risk. People with serious debt problems and in severe economic difficulties experience psychological distress [13], poor self-reported health [14], mental ill health or mental disorders and particularly depression [2, 3, 7, 8, 15–19], sleep problems [20, 21], chronic diseases such as hypertension or diabetes [2, 7], overweight and obesity [4] and back pain or pain in general [5, 22]. Over-indebted individuals also show higher risks of self-destructive behaviors such as suicidal behavior [8, 19], smoking [6, 23], alcohol and drug abuse [18, 19] or problem gambling [18, 24].

Regardless of such evidence in the research literature, there is not a single study available on health effects or correlates of over-indebtedness in Switzerland, with the exception of an own recently published study focusing on traditional debt parameters and the feeling of loosing or having no control over life (loss of sense of control and mastery) as explanatory or risk factors of the (poor) mental health of over-indebted [25]. The present population-based cross-sectional study therefore intends to fill the existing research gap and to be the first study of its kind in Switzerland to examine prevalence rates of the most common general, physical, mental and psychosomatic health problems among over-indebted people compared to the general population and to estimate the relative health risks of over-indebtedness. It is expected, of course, that these prevalence rates and relative risks are significantly and substantially increased among over-indebted individuals compared to the standard or reference population of mostly not over-indebted people. And it can be assumed finally that in a rich country like Switzerland where over-indebtedness (just like poverty) is a marginal and largely tabooed phenomenon and where there is no official debt relief procedure and therefore practically no way out of the financial misery and private insolvency, i.e. almost no possibility to leave the debts completely behind one day, over-indebtedness may be particularly detrimental to health and even more strongly associated with health problems than elsewhere. Regarding the socio-demographic characteristics of over-indebted individuals and based on evidence and findings from previous international studies [4, 21, 22], it is further expected that these people are comparably young, have lower educational attainment, have lower income, and are more often unmarried and unemployed.

Against this background, the objective of this explorative study was to examine the following research questions:

1. How can over-indebted people in Switzerland be characterized socio-demographically and socio-economically and in what way do they differ on average from the general population?

2. Do over-indebted individuals have a comparably bad health status? What are the prevalence rates of selected and specific poor general, physical and/or mental health outcomes among over-indebted people compared to the general population?

3. Is over-indebtedness a major and independent health risk factor regardless of other health-related personal characteristics, proven risk factors and possible confounders such as female sex, older age, foreign nationality, little education, low income, unemployment etc.?

## Methods

### Data and study population

Given the fact that data on over-indebtedness are difficult to capture and to collect and are usually not assessed in health-related population surveys in Switzerland and therefore largely missing, own data were needed and provided to study the financial situation and the health status of over-indebted individuals. These self-collected survey data were used for the present study and linked with previously collected secondary data from a nationally representative survey in order to compare over-indebted individuals with not overly indebted ones with regard to their (mental) health status. More precisely, the study sample consisted of 219 adults living in the Canton of Zurich who are seeking debt advice and a way out of their financial hardship and misery, and 1,997 representatives of the general population of the same age and residential region (see Fig 1).

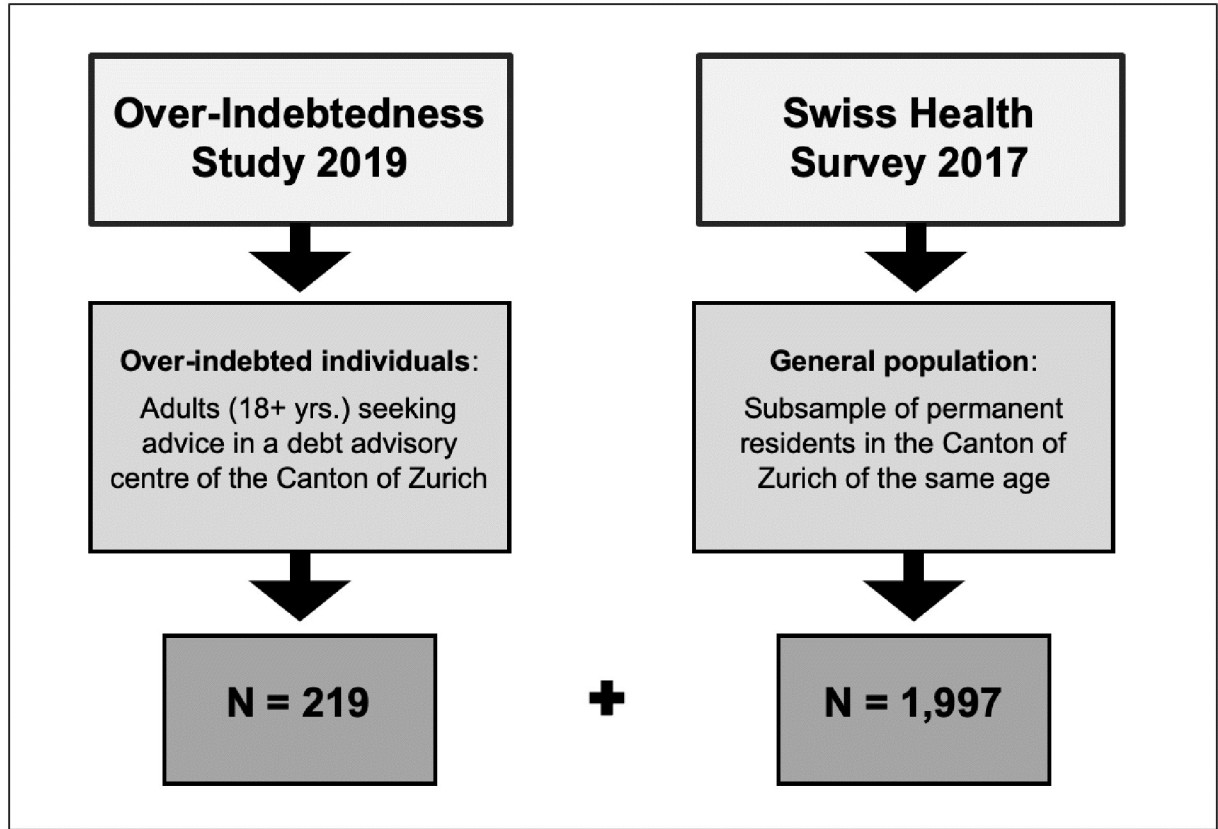

**Fig 1. The study population: An aggregation and combination of two randomly selected samples of over-indebted individuals and the general population in the Canton of Zurich.**

The 219 overly indebted individuals were surveyed in 2019 and recruited among clients of all four official debt advisory centers in the Canton of Zurich who are by definition, from experience of debt advisors and from their own point of view unexceptionally over-indebted. The 1,997 randomly selected survey participants representing the general population were questioned within the nationally representative Swiss Health Survey of 2017 (complete subsample of survey participants aged 18 and older and living in the Canton of Zurich).

In the self-developed questionnaire for the survey among over-indebted clients of debt advisory centers a core set of questions was used and taken from the Swiss Health Survey in order to have identical measures and variables to allow a data linkage between the two surveys and a direct comparison of the two survey populations which was intended right from the beginning.

In total, the aggregated or combined dataset and study sample covered 2,216 adults living in the Canton of Zurich (at the time of data collection), with a share of roughly 10% of 'exposed' or rather over-indebted individuals and 90% of 'non-exposed', i.e. completely or predominantly not over-indebted persons as the comparison or reference group.

This kind of linkage and combination of different cross-sectional data from (a) a target group-specific data collection and (b) a nationally representative survey, and the pooling of two random samples from a collective of over-indebted individuals and the general population (as the reference group) as illustrated in Fig 1 was also done in a few previous studies in Germany [4, 5, 21].

According to experiences and reports from the debt advisors of these centers unexceptionally all advice seeking clients are unable to meet their financial obligations for mostly a longer period of time and see it as a burden. Reports of debt advisors further suggest that the longer and the more highly these individuals are already indebted the more burdensome they find it and the more likely they seek advice from a debt advisory center. This indicates that those included in the study population constitute really the tip of the iceberg of (over-)indebtedness and not just a random selection of only lightly and temporarily indebted individuals.

## Ethical approval

The study was granted exemption from requiring ethics approval, because the study does not fall within the scope of the Human Research Act (HRA). (Kanton Zürich, Kantonale Ethikkommission. BASEC-Nr. Req-2019-00173).

## Measures

The only explaining, predicting or exposure variable that was used and studied here was the following independent and dichotomous variable:

**Over-indebtedness.** In this study over-indebtedness as the assumed main predictor or risk factor of poor health outcomes was defined and measured simply as seeking advice from a debt advisory center and being selected and asked for participation by a debt advisor which indicates serious financial difficulties. Participating in the survey at the debt advisory centers was categorized as being over-indebted ("exposed"). Participating in the Swiss Health Survey was considered as representing the general population and therefore as not or only marginally being over-indebted on a group level ("non-exposed"). This means that controls in this epidemiological study most probably are over-indebted only in a low and statistically negligible number of cases, if at all.

The following two general or physical and two mental health outcomes were studied (as dependent variables) in association with over-indebtedness:

**Self-rated health (SRH).**   SRH as a widely used and well-established general health indicator and a proven strong predictor of mortality was measured as is usual by a single item on the respondents' self-assessment of their state of health, with response options from 1 (*very good*) to 5 (*very bad*). A self-rating of health as less than good, i.e. only 3 (*moderate*) or even 4 (*bad*) or 5 (*very bad*), was categorized as poor SRH.

**Musculoskeletal disorders (MSDs).**   MSDs were assessed by combined self-reports of back pain or low back pain and neck or shoulder pain. I respondents had experienced both of these two types of complaints or health symptoms within the last 4 weeks and at least one of them on an increased level ('strong pain'), MSDs were classified as severe.

**Depression.**   Depression was measured by the 9-item depression scale of the Patient Health Questionnaire (PHQ-9), an established and highly reliable screening instrument for depression. Respondents were asked about being impaired in the last 2 weeks by complaints, attitudes, and emotions like joylessness, indifference, despair, melancholy, fatigue, loss of appetite, restlessness, poor concentration, death wish and so on. Response categories for all of the nine items or symptoms were 0 (*not at all*), 1 (*on single days*), 2 (*on more than half of the days*), and 3 (*almost every day*). The sum or total score was classified into different degrees of severity: total score 0–4 (no or minimal depression), 5–9 (mild depression), 10–14 (moderate depression), 15–19 (moderately severe depression), 20–27 (severe depression). Internal consistency of the 9-item depression scale was fairly high (Cronbach's alpha = .88).

**Sleep disorders (SDs).**   SDs were assessed by a single item asking if the respondent had had any difficulties falling or staying asleep in the last 4 weeks, with response options from 0 (*not at all*), 1 (*a little*), 2 (*severe*).

The following socio-demographic and socio-economic characteristics were additionally assessed and used as control variables (covariates) and for statistical adjustments:

Control variables were equally measured in both surveys by using identical questions in the over-indebtedness study than in the Swiss Health Survey. Respondents of both surveys were directly asked about their *sex*, *age*, *nationality*, their actual *marital* and *employment status* and about their highest *educational level* achieved so far and their personal *monthly net income* (excluding social security and pension contributions and including wages, alimonies etc.).

## Analyses

To address the first research question on the specific and possibly different characteristics of over-indebted individuals compared to the general population, relative frequencies (percentages) were calculated for different socio-demographic and socio-economic characteristics (sex, age, nationality, marital status, education, income, employment status) and for the study population of over-indebted individuals (exposed group) and the general population (reference or non-exposed group) separately.

In order to respond to the second research question about a potentially and comparably poor health status of over-indebted people, simple prevalence rates were calculated for all studied poor health outcomes (poor self-rated health, severe musculoskeletal disorders, moderate to severe depression, severe sleep disorders), again for both populations or study samples separately.

Finally, multivariate logistic regression analyses were performed and multiple-adjusted odds ratios (aOR) were calculated in order to answer the third research question and to estimate the independent effect of the main exposure or predictor variable (over-indebtedness) on the selected poor health outcomes with simultaneous consideration of all other risk and/or confounding factors (covariates).

## Results

Regardless of the personal characteristics of the over-indebted (research question 1), their health status (research question 2) in comparison with the general population and its statistically calculated association with over-indebtedness (research question 3) over-indebtedness is attributed to different causes by the affected persons themselves. When asked about the single or multiple reasons for their (over-)indebtedness out of a list of given 19 possible reasons clients of the debt advisory centers who participated in the survey most frequently stated unemployment (38%), problems with managing the finances (35%), illness or accident (29%), separation or divorce (27%) and debts made by others (22%), followed by other reasons (19%), low income (13%) and high fixed costs (12%). More than a quarter of the respondents and over-indebted individuals do not expect to ever be able to repay their debts completely. Another third is not expecting to repay them within the next five years.

The aggregated study sample and pooled data included 2,216 individuals, whereby roughly 10% (N = 219) of the sample were seeking advice from a debt advisory center and were therefore categorized as over-indebted and the other 90% (N = 1,997) were respondents of the Swiss Health Survey and representatives of the general population and considered as being not over-indebted (reference group).

The two groups differed considerably from each other with respect to age, nationality, and educational attainment, and marital and employment status (see Table 1). Over-indebted individuals were, compared with their not over-indebted counterparts of the general population, younger and most likely in their 30s or 40s (56% vs. 38%), mostly not highly educated (73% vs. 48%), had predominantly a low or middle personal income (72% vs. 50%), were quite frequently of foreign nationality (32% vs. 24%), were mostly unmarried, separated or divorced (77% vs. 49%), and were much more likely to be unemployed (14% vs. 3%).

In addition to these socio-demographic and socio-economic differences between the two populations or subsamples, much higher prevalence rates of poor health outcomes among the over-indebted individuals compared to the general population in the Canton of Zurich were observed. Nearly 53% of the over-indebted subsample but only 14% of the predominantly not over-indebted general population showed poor SRH (see Table 2). SRH is well-known as a valid general health indicator and good predictor of life expectancy. For other physical and mental health outcomes, this difference in the group comparison was even more pronounced, to the disadvantage of the over-indebted (see Table 2): severe musculoskeletal disorders (37% vs. 8%), moderate to severe depression (54% vs. 7%) and severe sleep disorders (40% vs. 6%). In sum, over-indebted individuals report much more often a poor health status compared to the general population.

In multiple logistic regression analyses (see Tables 3 and 4), over-indebtedness as the main predictor turned out to be the most strongest risk or explanatory factor of all poor health outcomes studied (aOR = 8.5–11.6), even after adjustment for numerous control variables. Besides this main finding, remarkable health differences associated with sociodemographic characteristics were observed: Men showed mostly a significantly lower health risk than women (aOR = 0.5–0.6). With increasing age, poor SRH and severe SDs were significantly more likely, whereas depression was less common, independent of over-indebtedness or no over-indebtedness. With decreasing educational attainment, the likelihood and risk of poor general health (moderate to very bad SRH: from 10% to 36%, aOR = 2.7), mental health (moderate to severe depression: from 7% to 23%, aOR = 2.7), psychosomatic health (severe SDs: from 6% to 17%, aOR = 1.8), and physical health (severe MSDs: from 6% to 19%, aOR = 2.6) increased gradually and substantially. Low or no personal income was associated with a significantly increased probability of and risk for poor SRH, independent of educational attainent or

**Table 1. Socio-demographic characteristics of the study population, stratified by subsamples (N = 2,216).**

| | | Study population | |
|---|---|---|---|
| | | Clients of debt advisory centers (N = 219)[a] | Representatives of the general population (N = 1,997)[b] |
| **Sex** | Men | 52.8% | 50.5% |
| | Women | 47.2% | 49.5% |
| **Age** | 18–30 years | 20.8% | 17.7% |
| | 31–40 years | 29.3% | 19.2% |
| | 41–50 years | 26.4% | 18.5% |
| | 51–60 years | 17.1% | 16.0% |
| | 61–70 years | 3.7% | 13.5% |
| | 71+ years | 2.8% | 15.1% |
| **Nationality** | Swiss (incl. dual citizenship) | 68.5% | 75.9% |
| | Foreign | 31.5% | 24.1% |
| **Marital status** | Married (2, 6) | 22.7% | 51.1% |
| | Single (unmarried) / separated (1, 5) | 53.2% | 34.0% |
| | Divorced / widowed (3, 4, 7) | 24.1% | 14.9% |
| **Educational attainment**[c] | Low (0–1) | 16.2% | 15.1% |
| | Medium (2) | 56.5% | 32.9% |
| | High (3–5) | 15.8% | 21.3% |
| | Very high (6–7) | 11.6% | 30.7% |
| **Income status**[d] | No income | 2.7% | 8.0% |
| | Low (CHF 3,000 or below) | 26.0% | 24.4% |
| | Medium (CHF 3,001–6,000) | 46.1% | 26.0% |
| | High (CHF 6,001–9,000) | 20.5% | 27.0% |
| | Very high (above CHF 9,000) | 4.6% | 14.6% |
| **Employment status** | Employed | 67.6% | 71.6% |
| | Unemployed (jobless) | 13.7% | 2.5% |
| | Non-working | 18.7% | 25.9% |

[a]Over-indebted adults seeking advice from one of the official debt advisory centers in the Canton of Zurich and participating in the over-indebtedness study (and survey) of 2019

[b]Subsample of the nationally respresentative Swiss Health Survey of 2017, restricted to adult respondents (aged 18 yrs. and older) from the Canton of Zurich; weighted data

[c]Educational attainment (highest level achieved): *low* (no or only compulsory education), *medium* (basic vocational education), *high* (higher vocational education), *very high* (university degree)

[d]Personal net income per month

of overindebtedness or no over-indebtedness. However, there was no a linear and/or significant effect of the amount of income on the other health outcomes examined. Foreign nationality was an independent risk factor for at least poor SRH and moderate to severe depression (aOR = 1.5 in each case). And finally, being unemployed was associated with a comparably high probability of and relative risk for poor SRH (37% vs. 18%, aOR = 1.7) and moderate to severe depression (32% vs. 11%, aOR = 1.6), even though multiple adjusted association measures or odds ratios (aOR) were not statistically significant due to comparably low numbers and large confidence intervals.

## Discussion

Over-indebtedness and possible health consequences among affected persons is clearly an unsufficiently studied topic and population in health-related research, not only but particularly

**Table 2. Health-related characteristics of the study population, stratified by subsamples (N = 2,216).**

| | | Study population | |
|---|---|---|---|
| | | **Clients of debt advisory centers (N = 219)[a]** | **Representatives of the general population (N = 1,997)[b]** |
| **Self-rated health** | Very good (1) | 11.5% | 40.6% |
| | Good (2) | 35.8% | 45.2% |
| | Moderate (3) | 32.6% | 12.0% |
| | Bad to very bad (4–5) | 20.2% | 2.3% |
| **Musculoskeletal disorders** | No (2) | 18.6% | 42.7% |
| | Moderate (3–4) | 44.7% | 49.7% |
| | Severe (5–6) | 36.7% | 7.7% |
| **Depression** (depressive symptoms) | No or minimal (0–4) | 19.4% | 67.7% |
| | Mild (5–9) | 26.7% | 25.4% |
| | Moderate (10–14) | 30.0% | 4.5% |
| | (Moderately) severe (15–27) | 24.0% | 2.4% |
| **Sleep disorders** | Not at all (1) | 24.0% | 65.4% |
| | A little (2) | 36.4% | 28.5% |
| | Severe (3) | 39.6% | 6.1% |

[a]Over-indebted adults seeking advice from one of the official debt advisory centers in the Canton of Zurich and participating in the over-indebtedness study (and survey) of 2019

[b]Subsample of the nationally respresentative Swiss Health Survey of 2017, restricted to adult respondents (aged 18 yrs. and older) from the Canton of Zurich; weighted data

for Switzerland where population-based self-reported data from over-indebted individuals were completely lacking so far. In order to fill this data and research gap and to compare over-indebted individuals with the general population regarding their health, we collected own survey data among over-indebted individuals which are not only much understudied but also constantly and strongly underrepresented in population-based health-related surveys.

Our research interest was to examine if, how, and how much over-indebted individuals on average differ from others in their social and health status. The study findings reveal that they do differ in various respects and to a great extent. The over-indebted individuals in our study are not only younger, definitely less educated, much more often unmarried and unemployed, and have lower earned income, but also are at much higher risk for poor health—general, physical, and mental health. Multivariate association analyses or, more precisely multiple logistic regression analyses, have clearly shown that the relatively poor health status of the over-indebtedness group compared to the general population cannot be explained by their different demographic composition (significantly higher proportions of younger age groups, of foreigners, and of unmarried, separated, or divorced individuals), at least not fully. And in particular the poor health status of over-indebted individuals cannot be attributed to their comparably low social status (substantially higher rate of unemployment and lower proportions of people with high educational attainment and high earned income), as could have been expected. Independent of, or rather adjusted for, these characteristics, relative risks of poor self-rated health, severe musculoskeletal disorders and sleep disorders and moderate to major depression are greatly increased and turn out to be 8-fold up to 11-fold higher among the over-indebted individuals compared to average people in the general population.

These health risks of people with over-indebtedness in the Canton or Zurich seem to be far greater and much more pronounced than have been found in the very few existing and only

**Table 3. Association of over-indebtedness and covariates with physical health problems (N = 2,216).**

| | Poor self-rated health (SRH) (3–5) | | | Severe musculoskeletal disorders (MSDs) (5–6) | | |
|---|---|---|---|---|---|---|
| | % | aOR | 95% CI | % | aOR | 95% CI |
| **Total study population** | **18.5** | | | **10.4** | | |
| **Over-indebtedness** (client of debt advisory center) | | | | | | |
| No | 14.8 | 1 | | 7.6 | 1 | |
| Yes | 52.8 | 9.30*** | 6.46–13.41 | 36.7 | 8.51*** | 5.74–12.62 |
| **Sex** | | | | | | |
| Female | 20.0 | 1 | | 13.2 | 1 | |
| Male | 16.8 | 0.89 | 0.69–1.14 | 7.3 | 0.47*** | 0.34–0.65 |
| **Age** | | | | | | |
| 18–30 years (1–2) | 10.0 | 1 | | 9.4 | 1 | |
| 31–50 years (3–4) | 14.4 | 1.87** | 1.16–3.03 | 10.3 | 1.12 | 0.68–1.84 |
| 51–70 years (5–6) | 22.9 | 4.70*** | 2.86–7.75 | 11.4 | 1.53 | 0.90–2.61 |
| 71+ years (7–8) | 28.1 | 6.57*** | 3.81–11.33 | 9.3 | 1.39 | 0.74–2.62 |
| **Nationality** | | | | | | |
| Swiss (incl. dual citizenship) | 17.4 | 1 | | 10.2 | 1 | |
| Foreign | 22.1 | 1.53** | 1.14–2.04 | 11.1 | 1.00 | 0.70–1.42 |
| **Civil / marital status** | | | | | | |
| Married (2, 6) | 18.1 | 1 | | 9.5 | 1 | |
| Single / separated (1, 5) | 14.3 | 0.88 | 0.63–1.23 | 9.9 | 0.78 | 0.52–1.15 |
| Divorced / widowed (3, 4, 7) | 27.8 | 1.10 | 0.81–1.51 | 13.9 | 0.93 | 0.62–1.39 |
| **Educational attainment** | | | | | | |
| Low (0–1) | 36.3 | 2.71*** | 1.77–4.14 | 18.9 | 2.57*** | 1.51–4.35 |
| Medium (2) | 22.1 | 1.51* | 1.08–2.12 | 12.3 | 1.50 | 0.98–2.29 |
| High (3–5) | 16.0 | 1.54* | 1.05–2.27 | 9.6 | 1.53 | 0.95–2.46 |
| Very high (6–7) | 9.9 | 1 | | 5.8 | 1 | |
| **Income** (personal monthly net income) | | | | | | |
| *Don't know / no answer* | 15.9 | 1.30 | 0.76–2.23 | 5.8 | 0.70 | 0.33–1.52 |
| No income | 19.1 | 2.04* | 1.17–3.55 | 11.2 | 1.33 | 0.68–2.60 |
| Low (< = CHF 3,000) | 27.8 | 2.23*** | 1.48–3.36 | 13.1 | 1.09 | 0.66–1.81 |
| Medium (CHF 3,001–6,000) | 18.9 | 1.36 | 0.93–1.98 | 12.0 | 1.09 | 0.69–1.71 |
| High very high (> CHF 6,000) | 10.1 | 1 | | 6.6 | 1 | |
| **Employment status** | | | | | | |
| Employed / not working | 17.9 | 1 | | 10.2 | 1 | |
| Unemployed (registered) | 36.5 | 1.72 | 0.94–3.15 | 15.1 | 0.71 | 0.34–1.50 |
| Number of cases in model | | 2,205 | | | 2,203 | |

*p≤.05;

**p < .01;

***p < .001

partly comparable cross-sectional studies in other European countries such as Germany, England or Sweden [3, 18, 21]. There is just one previous population-based study from Germany [5] that found similarly strong or even stronger effects of over-indebtedness on the occurrence of back pain (aOR = 10.9) than this study found for severe musculoskeletal disorders, i.e. severe (low) back pain combined with neck and shoulder pain (aOR = 8.5).

However, since the indicators used for possibly the same health outcomes were not identical across the studies, and/or categorizations or severities of these indicators were usually

**Table 4. Association of over-indebtedness and covariates with mental health problems (N = 2,216).**

| | Moderate to severe depression (10–27) | | | Strong sleep disorders (3) | | |
|---|---|---|---|---|---|---|
| | % | aOR | 95% CI | % | aOR | 95% CI |
| **Total study population** | **11.8** | | | **9.8** | | |
| **Over-indebtedness** (client of debt advisory center) | | | | | | |
| No | 7.1 | 1 | | 6.6 | 1 | |
| Yes | 53.9 | 11.59*** | 8.01–16.77 | 39.6 | 9.85*** | 6.65–14.58 |
| **Sex** | | | | | | |
| Female | 13.1 | 1 | | 11.9 | 1 | |
| Male | 10.2 | 0.61** | 0.45–0.84 | 7.5 | 0.55*** | 0.40–0.76 |
| **Age** | | | | | | |
| 18–30 years (1–2) | 17.3 | 1 | | 7.9 | 1 | |
| 31–50 years (3–4) | 13.7 | 0.77 | 0.49–1.18 | 10.7 | 1.70* | 1.01–2.88 |
| 51–70 years (5–6) | 10.0 | 0.79 | 0.48–1.30 | 10.9 | 2.30** | 1.30–4.05 |
| 71+ years (7–8) | 4.2 | 0.40* | 0.19–0.83 | 7.2 | 1.72 | 0.86–3.42 |
| **Nationality** | | | | | | |
| Swiss (incl. dual citizenship) | 10.1 | 1 | | 9.6 | 1 | |
| Foreign | 16.8 | 1.49* | 1.06–2.09 | 10.6 | 0.94 | 0.65–1.36 |
| **Civil / marital status** | | | | | | |
| Married (2, 6) | 7.5 | 1 | | 7.4 | 1 | |
| Single / separated (1, 5) | 16.7 | 1.51* | 1.04–2.19 | 10.9 | 1.23 | 0.83–1.83 |
| Divorced / widowed (3, 4, 7) | 14.9 | 1.48 | 0.95–2.30 | 15.3 | 1.47 | 0.97–2.22 |
| **Educational attainment** | | | | | | |
| Low (0–1) | 23.1 | 2.70*** | 1.59–4.58 | 16.7 | 1.83* | 1.06–3.15 |
| Medium (2) | 15.5 | 1.38 | 0.91–2.11 | 10.6 | 1.05 | 0.68–1.61 |
| High (3–5) | 10.7 | 1.34 | 0.84–2.15 | 10.3 | 1.50 | 0.94–2.40 |
| Very high (6–7) | 6.7 | 1 | | 6.2 | 1 | |
| **Income** (personal monthly net income) | | | | | | |
| No income | 8.1 | 0.63 | 0.29–1.37 | 7.9 | 1.12 | 0.53–2.38 |
| Low (< = CHF 3,000) | 15.3 | 1.01 | 0.61–1.67 | 13.3 | 1.45 | 0.87–2.40 |
| Medium (CHF 3,001–6,000) | 14.6 | 0.92 | 0.59–1.43 | 11.3 | 1.07 | 0.67–1.68 |
| High very high (> CHF 6,000) | 7.9 | 1 | | 6.6 | 1 | |
| *Don't know / no answer* | 3.7 | 0.52 | 0.21–1.28 | 4.1 | 0.61 | 0.26–1.44 |
| **Employment status** | | | | | | |
| Employed / non-working | 11.1 | 1 | | 9.4 | 1 | |
| Unemployed (registered) | 32.4 | 1.57 | 0.82–3.02 | 20.3 | 1.09 | 0.55–2.19 |
| Number of cases in model | | 2,153 | | | 2,204 | |

*p≤.05;

**p < .01;

***p < .001

different, the studies' findings, or rather the strength of the associations found between over-indebtedness and particular health outcomes as binary exposure and outcome variables, cannot actually be compared. This is all the more so because the often-used odds ratios, as measures of association and proxies for the relative risk, are not standardized and therefore are not comparable statistical figures, anyway. Therefore, there is in fact no other international study that can be compared with or used in support of the present study. Nevertheless, the findings of this and previous or recent studies in other and 'comparable' European countries (Germany,

Sweden, Finland, England or the United Kingdom) go in the same direction and show consistently a clear and strong negative relationship between over-indebtedness and health.

## Limitations

Since cross-sectional data were used for this study, causal inferences cannot be drawn. It remains unclear in this study if the burden of over-indebtedness sooner or later impairs an individual's health or if an impaired health in the long run leads to financial difficulties due to a higher risk of unemployment or working poor and getting a low income job. The assumed causation behind the strong association found is possibly reversed or—even more plausibly—bi-directional.

As no or only insufficient information about the sustainability of the participants' debt situation is provided or assessed in the Swiss Health Survey, a misclassification of survey participants into exposed individuals and non-exposed individuals cannot be completely excluded. It may be reasonably assumed that at least a few over-indebted individuals will also be found in a random sample of the general resident population, although they are most likely relatively few in number and strongly underrepresented. In other words: An information or misclassification bias may have occurred in this study. However, as such a measurement error or misclassification of the exposure status is random—i.e. non-differential, insofar as the error does not differ systematically between the cases and the controls and is independent of the disease status, the potential systematic bias is predictable and goes towards the null value and therefore may result in an underestimation of the risk and the true strength of association between exposure and outcome or disease.

Considering that the most (objective) indicators individually are not capable to completely capture over-indebtedness [12], probably the best method to measure over-indebtedness in an observational study is to ask people directly whether or not they are facing debt repayment difficulties [12]. But asking such delicate questions on a most sensitive issue like over-indebtedness is fairly problematic. As a result, over-indebted individuals and particularly clients of debt advisory centres which are anyway low in number tend to be underrepresented in population-based surveys. The most exposed and affected among them may be out of reach, unavailable or unable or simply not willing to admit and expose themselves and to participate in population surveys. In other words: Studying over-indebted and thus stigmatized and marginalized individuals carries the risk of a self-selection or participation bias (non-response bias) since the reasons of non-responders to self-exclude from the study or to refuse to participate in the survey might be systematically correlated with the outcomes under study. This undermines the external validity of the study results (non-respresentative findings) but again presumably leads to an underestimation rather than an overestimation of the "true" effects.

## Conclusion

This study, as the only Swiss study that has ever been conducted or published on this topic, can be understood as indicating that there is a particularly strong negative association between over-indebtedness and health in a rich country like Switzerland, where over-indebtedness is—or is supposed to be—a marginal phenomenon and where, hence, there is only very limited possibility for debt relief. What is more burdensome and detrimental to health seem to be great financial difficulties that lead to over-indebtedness and, consequently, an abrupt and/or complete loss of social status. Therefore, from a public health point of view, one of the best prevention measures could be to give the individuals concerned prospects for debt relief as soon as possible and thus the possibility of a fresh start.

## Supporting information

**S1 Data.**
(XLSX)

**S1 Questionnaire.**
(PDF)

## Author Contributions

**Conceptualization:** Oliver Hämmig.

**Data curation:** Oliver Hämmig, Joanna Herzig.

**Formal analysis:** Oliver Hämmig.

**Investigation:** Oliver Hämmig, Joanna Herzig.

**Methodology:** Oliver Hämmig.

**Project administration:** Joanna Herzig.

**Writing – original draft:** Oliver Hämmig.

**Writing – review & editing:** Oliver Hämmig, Joanna Herzig.

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
