## [Decision Letter · Decision Letter 0]

20 Jan 2022

PONE-D-21-26584Over-indebtedness and health in Switzerland: a cross-sectional study comparing over-indebted individuals and the general populationPLOS ONE

Dear Dr. Hämmig,

Thank you for submitting your manuscript to PLOS ONE. After careful consideration, we feel that it has merit but does not fully meet PLOS ONE’s publication criteria as it currently stands. Therefore, we invite you to submit a revised version of the manuscript that addresses the points raised during the review process.

 Your paper has been reviewed by two expert referees, who (although highlighting the value and merit of the work) request for some major changes, amendments and clarifications from you. After a careful review of the manuscript, I agree with them on the fact that the paper must be improved, especially in regards to methodological flaws that still can be fixed in order to better support and provide validity to your conclusions, even in a post-study phase.Also, please note the need to strengthen the discussion of the paper. Personally, I would suggest you to check the linearity and raise the discussion and conclusions in the light of the core study aim.

We look forward to receiving your revised manuscript.

Kind regards,

Sergio A. Useche, Ph.D.

Academic Editor

PLOS ONE

https://journals.plos.org/plosone/s/file?id=ba62/PLOSOne_formatting_sample_title_authors_affiliations.pdf”

Reviewers' comments:

Reviewer's Responses to Questions

**Comments to the Author**

1. Is the manuscript technically sound, and do the data support the conclusions?

Reviewer #1: Partly

Reviewer #2: Yes

2. Has the statistical analysis been performed appropriately and rigorously? 

Reviewer #1: I Don't Know

Reviewer #2: Yes

3. Have the authors made all data underlying the findings in their manuscript fully available?

Reviewer #1: No

Reviewer #2: No

4. Is the manuscript presented in an intelligible fashion and written in standard English?

Reviewer #1: No

Reviewer #2: Yes

5. Review Comments to the Author

Reviewer #1: Summary

The paper reports on the association between over-indebtedness and health, based on cross-sectional studies conducted in Switzerland, a) survey among over-indebted clients of debt advisory centers in the Canton of Zurich (over-indebtedness study, n=219) and b) nationally representative Swiss Health Survey (n=1997). The combined dataset included n=2216 adults aged 18 and over. Clients of debt advisory centers (a) were considered over-indebted, whereas respondents of the Swiss Health Survey (b) were considered not over-indebted. The paper described a higher prevalence of poor self-rated health, musculoskeletal disorders, depression and sleep disorders among over-indebted individuals compared to the Swiss general population. Multiple logistic regression analyses showed a significant association between over-indebtedness of these self-reported physical and mental health outcomes, adjusted for sex, age, education, nationality, marital status, employment status.

From a public health perspective, the research question is highly relevant and so far, little to no data is available on the health status of over-indebted individuals, specifically in Switzerland. From my point of view, however, there are several major issues that require revision. In the following, I focus on these major issues and may provide more comments on further minor issues at a later stage.

(1) Methods. The authors combined cross-sectional data among over-indebted clients of debt advisory centers collected in 2019, with the Swiss Health Survey conducted in 2017.

a. Please clarify the recruitment process and response rate in both surveys. Did both surveys include only respondents from the Canton of Zurich? If not, to what extent are the four official debt advisory centers in the Canton of Zurich and clients presenting to these centers representative for centers and clients across Switzerland? Please clarify the study population early in the methods section. What eligibility criteria was applied in the Swiss Health Survey? What were sources and methods of selection of participants in the Swiss Health Survey? How was the subsample of adult respondents of the Swiss Health Survey selected? In Table 1 the authors mention that the Swiss Health Survey data was weighted. How was the weighting process carried out?

b. In the methods section, the authors state that descriptive statistics, bivariate and multivariate association analyses were performed (266-274). From my point of view, the authors should revise this section to clarify what measures they used to achieve which purpose. The authors state “multiple adjusted odds ratios (aOR) as measures of the relative risk” (273) which, in my view, is not a correct description of the statistical measure. What significance level was applied? How were missing data addressed? The different numbers of cases per model (Table 3, Table 4) indicate that individual cases were excluded from the multiple logistic regression analyses. Did the authors conduct any sensitivity analyses?

c. In order to clarify the methodological approach, I suggest to present key elements of the study design early in the paper, and to provide detailed information later. From my perspective, the authors could focus more on describing key characteristics of the methodological approach (study design, setting, participants, variables, study size, variables, statistical methods) and reduce additional details (e.g. 207-214) to avoid redundancy (“As already mentioned before…” 215) and, above all, make it easier for readers to understand which methods were used. The authors, for instance, list measures of over-indebtedness (178ff.) in the methods section and describe in detail why primary data was collected (172-194). Instead, the authors could briefly summarize these aspects and elaborate on relevant aspects in the discussion section, e.g. with regards to potential limitations of the study design or measures used in the study.

(2) Results. In the results section, the authors report descriptive statistics and key findings of multiple regression analysis.

a. I suggest to revise the description of the characteristics of the sample included in the analyses. Please report numbers of individuals at each stage of the study, e.g. numbers potentially eligible, examined for eligibility, confirmed eligible and included in each survey, and numbers included in the combined dataset and numbers analysed.

b. Please also specify sociodemographic data (mean, standard deviation) for the subsamples (278-286). What were the proportions of individuals considered over-indebted and not over-indebted? Explicitly state the share of over-indebted individuals included in the combined dataset (278-280).

c. I recommend to revise the description of findings, specifically with regards to the link between income (income, categorized into no/ low / medium / high income) and health outcomes. The authors mention in the table footer (Table 4) that Odds ratios were “adjusted for sex, age, education, nationality, marital status, employment status” (342). (Why) did the authors not adjust for income? Moreover, the authors describe that “Low or no personal income was associated with a significantly increased probability of and risk for poor SRH, independent of educational attainent or of over-indebtedness or no over-indebtedness” (323-325). They also state “However, there was no a linear and/or significant effect of the amount of income on the other health outcomes examined.” (325-326). In my view, multiple logistic regression analysis is not suitable to examine a linear relationship (or probability) as stated by the authors.

d. From my point of view, male sex was not significantly associated with poor self-rated health but marked bold in Table 3 (aOR 0.89; CI 0.69-1.14).

(3) Background. Please clarify the objectives of the study. What are the key objectives of the study? Was the study conducted to compare sociodemographic characteristics of those over-indebted to the general population in the Canton of Zurich (166) and/or to study the prevalence of health problems and association between over-indebtedness and health (146-153)?

Overall, I recommend that authors have the manuscript proofread by native speakers to avoid misunderstandings. The designation of statistical methods should be critically examined in some places.

Checklists for reporting research results (such as STROBE) should be used by the authors to reduce the manuscript to key relevant reporting points, avoid redundancies and ensure that the procedure carried out is described transparently and comprehensibly.

Reviewer #2: Although this study is on interesting research topic, there is a main methodological concern and limitation which need to be addressed in this present study. This is related to definition of over-indebtedness construct which is dichotomous and lacking of indicators for over-indebtedness. It is important to explain why measuring over-indebtedness indicators were not given priority in designing this present study.

Besides that, in the introduction section—the author(s) may highlight why research addressing over-indebtedness is important in Switzerland and why over-indebtedness might be increase in Switzerland?

In line no 313--please revise and specify the statement “...main predictor turned out to be the strongest risk or explanatory factor by far…” Please specify IF the author(s) refer to “the poor health outcomes” or something else.

In line no 357-358--please revise and specify the regression analyses authors referring to—do the authors refer to multiple logistic regression?

6. PLOS authors have the option to publish the peer review history of their article (what does this mean?). If published, this will include your full peer review and any attached files.

Reviewer #1: No

Reviewer #2: No

---

## [Author Response · Author response to Decision Letter 0]

13 Apr 2022

We have now revised our manuscript entitled

Over-indebtedness and health in Switzerland: a cross-sectional study comparing over-indebted individuals and the general population

(PONE-D-21-26584)

Please find below the comments and points raised by the reviewers along with our replies. All corrections and additions in the manuscript are highlighted in yellow. The deleted words, sentences or paragraphs are crossed out.

Reviewer’ comments and authors’ replies 

Reviewer #1:

“Please clarify the recruitment process and response rate in both surveys. Did both surveys include only respondents from the Canton of Zurich? If not, to what extent are the four official debt advisory centers in the Canton of Zurich and clients presenting to these centers representative for centers and clients across Switzerland? Please clarify the study population early in the methods section. What eligibility criteria was applied in the Swiss Health Survey? What were sources and methods of selection of participants in the Swiss Health Survey? How was the subsample of adult respondents of the Swiss Health Survey selected? In Table 1 the authors mention that the Swiss Health Survey data was weighted. How was the weighting process carried out?” 

Clarifications on the data collection among over-indebted individuals have been made now (see yellow marked comments in the Methods section). In fact, we have completely reformulated and shortened the first part of the Methods section (“Data and study population”).

For further information about the nationally representative and publicly available data of the Swiss Health Survey and the sampling and weighting procedures see the following webpage and/or publication: 

• https://www.bfs.admin.ch/bfs/de/home/statistiken/gesundheit/erhebungen/sgb.html

• Federal Statistical Office FSO (2018). Die Schweizerische Gesundheitsbefragung 2017 in Kürze. Konzept, Methode, Durchführung. Bundesamt für Statistik BFS, Bern.

Background

“Please clarify the objectives of the study. What are the key objectives of the study? Was the study conducted to compare sociodemographic characteristics of those over-indebted to the general population in the Canton of Zurich (166) and/or to study the prevalence of health problems and association between over-indebtedness and health (146-153)?” 

In summarizing and clarifying the research interest of the present study three research questions to be addressed have now been explicitly formulated and inserted at the end of the Background section. 

Methods 

“In the methods section, the authors state that descriptive statistics, bivariate and multivariate association analyses were performed (266-274). From my point of view, the authors should revise this section to clarify what measures they used to achieve which purpose. The authors state “multiple adjusted odds ratios (aOR) as measures of the relative risk” (273) which, in my view, is not a correct description of the statistical measure. What significance level was applied? How were missing data addressed? The different numbers of cases per model (Table 3, Table 4) indicate that individual cases were excluded from the multiple logistic regression analyses. Did the authors conduct any sensitivity analyses?” 

We have reformulated almost the entire subsection “Analyses” in the Methods section accordingly. As regards some specific points raised by the reviewer we’d like to comment as follows:

• Odds ratios are commonly used and reported measures of association between exposures and outcomes and at least in case of rather uncommon or rare diseases or outcomes good proxy measures of the relative risk. However, the relative risk or risk ratio (RR) compares the risk of an event or disease in an exposed and a non-exposed group of individuals while the odds ratio (OR) compares the probability of an event or disease with the probability of the non-occurance of the event or disease. When an outcome occurs in less than 10% of the unexposed population the OR provides a reasonable approximation of the RR, but when an outcome is more common the OR overestimates the RR. Since OR and RR in fact are not exactly the same measures and since we calculated logistic regression analyses which do not provide RRs we have decided to forego comparing odds ratios with measures of relative risk.

• The significance level as usual was 5% (p<=.05), indicated in the Tables 3 and 4 as 95% confidence invervals. However, we have additionally provided higher significance levels of 1% and 1‰ (**p<.01, ***p<.001).

• There were only very few missing cases (less than 1%) except for one association analysis and health outcome (depression; less than 3%). Missing cases therefore were not addressed at all. Excluding these cases from the association analyses did not result in any significant loss of robustness or generalizability of the findings. 

• Sensitivity analyses were not performed since logistic regression analyses were performed and not linear regression analyses using standardized regression coefficients. However, group-specific prevalence rates were additionally provided and indicated in Tables 3 and 4 in order to better or more adequately evaluate the health risks (odds ratios) of the exposed (over-indebted) individuals, and to assess the influence of the covariates considered.

 “In order to clarify the methodological approach, I suggest to present key elements of the study design early in the paper, and to provide detailed information later. From my perspective, the authors could focus more on describing key characteristics of the methodological approach (study design, setting, participants, variables, study size, variables, statistical methods) and reduce additional details (e.g. 207-214) to avoid redundancy (“As already mentioned before…” 215) and, above all, make it easier for readers to understand which methods were used. The authors, for instance, list measures of over-indebtedness (178ff.) in the methods section and describe in detail why primary data was collected (172-194). Instead, the authors could briefly summarize these aspects and elaborate on relevant aspects in the discussion section, e.g. with regards to potential limitations of the study design or measures used in the study.” 

We fully agree with the reviewer’s suggestion and have strongly shortened, realigned and focused the Methods section (or rather the ‘Data and study population’ subsection). More specifically, we have completely deleted large parts of the subsection or moved them to the Discussion section (under ‘Limitations’), and have now inserted some newly formulated paragraphs. This has hopefully and effectively (in our opinion) much improved the understanding and/or transparency of the design, setting, size, participants and methods of the study – and avoided redundancy.

Results

“I suggest to revise the description of the characteristics of the sample included in the analyses. Please report numbers of individuals at each stage of the study, e.g. numbers potentially eligible, examined for eligibility, confirmed eligible and included in each survey, and numbers included in the combined dataset and numbers analysed,” 

There must be a misunderstanding here. There has not been a multistage recruitment or selection process or analysis procedure. We have simply put together two datasets and study samples, and have analyzed them subsequently as a whole. In one case the entire sample or survey population was used and in the other case a comparable subsample of a nationwide survey population was selected. And by the way, all numbers of individuals can be calculated from the relative frequencies (%) and the size of the (sub)populations (n/N) shown in Table 1. In our opinion it’s not essential or more informative to indicate relative and absolute frequencies. However, we have made some few adaptions and reformulations to hopefully avoid possible misunderstandings.

“Please also specify sociodemographic data (mean, standard deviation) for the subsamples (278-286). What were the proportions of individuals considered over-indebted and not over-indebted? Explicitly state the share of over-indebted individuals included in the combined dataset (278-280).” 

Do you refer to the characteristics provided in Table 1? In Table 1 for both subsamples relative frequencies (%) of socio-demographic characteristics are already provided. Means and standard deviations make little sense for categorical variables such as sex, nationality, marital status or employment status. And for ordinally scaled variables such as age, educational level or income they are less significant and little informative. And the share of the subsample of over-indebted individuals of the total study sample is noted (“roughly 10%”) and amounts exactly 9.9%. Absolute frequencies (numbers) of the two subsamples are provided in Table 1. 

To clarify once more: The subsample of 219 individuals comprises completely of over-indebted individuals whereas the reference group of the 1,997 respondents of the Swiss Health Survey are the representatives of the general population who are considered to be largely not overly indebted and therefore taken and treated as the reference group of the non-exposed, not over-indebted. This is now pointed out more clearly in the Methods section (subsection ‘Data and study population’).

“I recommend to revise the description of findings, specifically with regards to the link between income (income, categorized into no/ low / medium / high income) and health outcomes. The authors mention in the table footer (Table 4) that Odds ratios were “adjusted for sex, age, education, nationality, marital status, employment status” (342). (Why) did the authors not adjust for income? Moreover, the authors describe that “Low or no personal income was associated with a significantly increased probability of and risk for poor SRH, independent of educational attainent or of over-indebtedness or no over-indebtedness” (323-325). They also state “However, there was no a linear and/or significant effect of the amount of income on the other health outcomes examined.” (325-326). In my view, multiple logistic regression analysis is not suitable to examine a linear relationship (or probability) as stated by the authors.” 

You’re totally right regarding the footer. But of course all studied associations were adjusted for income. The whole footnote was simply an error and beyond that incomplete and unnecessary (and therefore only placed at the foot of Table 4 and not below Table 3 likewise). It was wrongly adopted from an old table…

Since all covariates or control variables for which the main predictor or risk factor (over-indebtedness) was adjusted were included in the logistic regression analyses and shown in Tables 3 and 4, such a footnote makes no sense at all. 

And by “linear” we meant a clear gradient or dose-effect relationship. We have reformulated the sentence accordingly.

But apart from that, we have chosen to use logistic regression analyses due to the limited level or scale of measurement (ordinal, NOT interval) and/or the inadequate marginal distribution (strongly skewed, NOT normal) of the outcome variables under study. Linear regression analyses in contrast were not suitable since they require interval scaled and normally distributed variables as dependent or outcome variables. We think anyway that logistic regression analyses are absolutely suitable to examine linear or rather dose-effect relationships, particularly when relationships are curve linear, when a gradient is expected and wants to be studied and estimated and/or when predictors (independent variables) are nominal scaled or were treated like categorical variables and, hence, an indicator method was used to contrast the different categories from an indicated reference category as we did.

“From my point of view, male sex was not significantly associated with poor self-rated health but marked bold in Table 3 (aOR 0.89; CI 0.69-1.14).” 

Yes, you are completely right. This is a mistake and has been corrected now. 

Reviewer #2:

“Although this study is on interesting research topic, there is a main methodological concern and limitation which need to be addressed in this present study. This is related to definition of over-indebtedness construct which is dichotomous and lacking of indicators for over-indebtedness. It is important to explain why measuring over-indebtedness indicators were not given priority in designing this present study.”

Two text passages have been inserted in the Discussion section (under the subsection “Limitations”) to address and discuss this concern as an additional limitation and to justify the chosen study design. 

“Besides that, in the introduction section—the author(s) may highlight why research addressing over-indebtedness is important in Switzerland and why over-indebtedness might be increase in Switzerland?” 

At the end of the Background section (just before the newly formulated research questions), we have now explicitly summarized what we tried to demonstrate all over the introduction, namely why research on this topic and population group is needed and important, particularly for Switzerland. 

“In line no 313--please revise and specify the statement “...main predictor turned out to be the strongest risk or explanatory factor by far…” Please specify IF the author(s) refer to “the poor health outcomes” or something else.” 

Yes, we have referred to Tables 3 and 4 as described and consequently to the four poor health outcomes shown in these tables. But we made this even more explicit now by specifying “of all poor health outcomes studied”. 

“In line no 357-358--please revise and specify the regression analyses authors referring to—do the authors refer to multiple logistic regression?” 

Yes, since multiple logistic regression analyses were the only multivariate statistical methods or regression analyses that we have used or made in this study. We have slightly reformulated the according sentence.

---

## [Decision Letter · Decision Letter 1]

15 Jun 2022

PONE-D-21-26584R1Over-indebtedness and health in Switzerland: a cross-sectional study comparing over-indebted individuals and the general populationPLOS ONE

Dear Dr. Hämmig,

Thank you for submitting your manuscript to PLOS ONE. After careful consideration, we feel that it has merit but does not fully meet PLOS ONE’s publication criteria as it currently stands. Therefore, we invite you to submit a revised version of the manuscript that addresses the points raised during the review process.

Your paper has been reassessed by two of our previous referees. Although our reviewers highlight the value of most of the improvements and rationales offered by you, more work seems to be needed. Specifically, many sections of the paper (especially those containing technical issues, results or their interpretation) remain unclear, requiring further major revisions from you, alongside a further round of reviews, in order to validate the adequacy of these amendments. Please submit your revised manuscript by Jul 30 2022 11:59PM. If you will need more time than this to complete your revisions, please reply to this message or contact the journal office at plosone@plos.org. Please include the following items when submitting your revised manuscript:A rebuttal letter that responds to each point raised by the academic editor and reviewer(s). You should upload this letter as a separate file labeled 'Response to Reviewers'.A marked-up copy of your manuscript that highlights changes made to the original version. You should upload this as a separate file labeled 'Revised Manuscript with Track Changes'.An unmarked version of your revised paper without tracked changes. You should upload this as a separate file labeled 'Manuscript'.

We look forward to receiving your revised manuscript.

Kind regards,

Sergio A. Useche, Ph.D.

Academic Editor

PLOS ONE

Reviewers' comments:

Reviewer's Responses to Questions

**Comments to the Author**

1. If the authors have adequately addressed your comments raised in a previous round of review and you feel that this manuscript is now acceptable for publication, you may indicate that here to bypass the “Comments to the Author” section, enter your conflict of interest statement in the “Confidential to Editor” section, and submit your "Accept" recommendation.

Reviewer #3: (No Response)

Reviewer #4: All comments have been addressed

2. Is the manuscript technically sound, and do the data support the conclusions?

Reviewer #3: Yes

Reviewer #4: Yes

3. Has the statistical analysis been performed appropriately and rigorously? 

Reviewer #3: Yes

Reviewer #4: Yes

4. Have the authors made all data underlying the findings in their manuscript fully available?

Reviewer #3: (No Response)

Reviewer #4: Yes

5. Is the manuscript presented in an intelligible fashion and written in standard English?

Reviewer #3: Yes

Reviewer #4: Yes

6. Review Comments to the Author

Reviewer #3: Important public health paper on over-indebtedness and health in Switzerland.

Paper is lengthy, particularly Introduction and Discussion. Try to short and avoid repetition.

Add paragraph on how the covariates were measured (e.g. income) or point to Table 3 and 4 (and add a footnote there for additional description).

The Analyses (line 269), first paragraph is stratified by sex for study population and second paragraph for over-indebted. That is unclear, and not only because in the Results the stratification took place for the over-indebted group only. More importantly, why this stratification and why (ultimately) only for the indebted group? Was there an interaction between sex and over-indebtedness?

Line 366: you indeed show that the covariates cannot (fully) explain the impact of over-indebtedness. The key thing is to add “fully”, as you have not shown mediation analyses indicating the extent to which the impact of over-indebtedness is explained by the covariates (but not fully, I agree).

Discussion, line 418-420. Why is the need for a longitudinal study related to the over-indebted people being hard to reach? Does it not fit better with the reversed causation problem (line 400)? 2nd: Talking about case – control study is confusing, as you seem to imply that the over-indebted are cases and the other are controls. In epidemiological research cases are the one with the disease and controls are the ones without disease. This is confusing (line 420-432). 3rd: paragraph 433-447 switches from a limitation (no variation within over-indebtness), which might or might not be problematic (as there might be differences in “the level of indebtedness”) to a report on a design that you did not use (having only over-indebted group) which is of course good, but one can always imagine more worse designs than your own. The whole paragraph is lengthy and too much words combining pros and cons in confusing way.

Minor

Reference for social disparities in Switzerland (line 154).

Methods: tip of the iceberg sentence (line 209): think of having such reflections in the Discussion.

Methods: reason for over indebtedness (line 211 and further): is that not more Results?

Type line 250.

Think of having a sentence in Measures saying you have two physical and two mental health outcomes. Otherwise, Table 3 and 4 initially are a bit unclear.

Line 31: better in Measures section on SRH?

Line 315-316: “very poor health status”? You do not know, you only know that they report poor health more often. Formulate otherwise.

Line 323: “by far”: uncommon in academic writings.

Line 381: aOR=8.5?

Reviewer #4: Dear Author,

The manuscript significantly improved after this revision.

However, introduction and limitation sections unnecessarily long that need to be concise before final decision.

7. PLOS authors have the option to publish the peer review history of their article (what does this mean?). If published, this will include your full peer review and any attached files.

Reviewer #3: **Yes: **Hans Bosma

Reviewer #4: No

---

## [Author Response · Author response to Decision Letter 1]

8 Aug 2022

Reviewer’ comments and authors’ replies 

Reviewer #3:

“Important public health paper on over-indebtedness and health in Switzerland.

Paper is lengthy, particularly Introduction and Discussion. Try to short and avoid repetition.”

We have shortened the Introduction section by about one third (more than one page), and in addition have reduced the text of the Discussion section substantially.

“Add paragraph on how the covariates were measured (e.g. income) or point to Table 3 and 4 (and add a footnote there for additional description).”

Done. A paragraph on the control variables was added at the end of the “Measures” section.

“The Analyses (line 269), first paragraph is stratified by sex for study population and second paragraph for over-indebted. That is unclear, and not only because in the Results the stratification took place for the over-indebted group only. More importantly, why this stratification and why (ultimately) only for the indebted group? Was there an interaction between sex and over-indebtedness?”

This stratification by sex for the over-indebted individuals only (in Tables 1 and 2) was done simply to better characterize the population of interest and because the two sexes differ greatly as regards their socio-demographic characteristics and their health status – and NOT due to any interaction between sex and over-indebtedness. However, these stratified descriptive analyses for the over-indebted group and not for the comparison group might be a bit confusing. And since the results do not respond basically to any of the research questions and therefore also were not further discussed in the Results or Discussion section we refrain from this stratification and changed the Tables 1 and 2 accordingly.

 “Line 366: you indeed show that the covariates cannot (fully) explain the impact of over-indebtedness. The key thing is to add “fully”, as you have not shown mediation analyses indicating the extent to which the impact of over-indebtedness is explained by the covariates (but not fully, I agree).”

We agree that some covariates in fact do have an explanatory or predictive effect but at least cannot fully explain the strong negative health impact of over-indebtedness. Therefore we have reformulated the according sentence as suggested. 

 “Discussion, line 418-420. Why is the need for a longitudinal study related to the over-indebted people being hard to reach? Does it not fit better with the reversed causation problem (line 400)?”

We have reformulated and strongly shortened the according paragraph.

 “2nd: Talking about case – control study is confusing, as you seem to imply that the over-indebted are cases and the other are controls. In epidemiological research cases are the one with the disease and controls are the ones without disease. This is confusing (line 420-432).”

We understand that this might be confusing and have reformulated the whole paragraph and cancelled the conclusion about alternative study designs of a prospective cohort or retrospective case-control study. 

 “3rd: paragraph 433-447 switches from a limitation (no variation within over-indebtness), which might or might not be problematic (as there might be differences in “the level of indebtedness”) to a report on a design that you did not use (having only over-indebted group) which is of course good, but one can always imagine more worse designs than your own. The whole paragraph is lengthy and too much words combining pros and cons in confusing way.”

We have now deleted the whole paragraph. 

Minor

• Reference for social disparities in Switzerland (line 154).

The entire paragraph has now been deleted in the course of the text cuts. 

• Methods: tip of the iceberg sentence (line 209): think of having such reflections in the Discussion.

No / no more (since the Discussion has partly shortened). 

• Methods: reason for over indebtedness (line 211 and further): is that not more Results?

Type line 250.

We have now moved the paragraph to the beginning of Results section and have put an introductory sentence in front.

• Think of having a sentence in Measures saying you have two physical and two mental health outcomes. Otherwise, Table 3 and 4 initially are a bit unclear.

Done. We have added a sentence on that under the Measures section (just below the description of the over-indebtedness variable).

• Line 31: better in Measures section on SRH?

???? (makes no sense; presumably wrong page reference?)

• Line 315-316: “very poor health status”? You do not know, you only know that they report poor health more often. Formulate otherwise.

Done.

• Line 323: “by far”: uncommon in academic writings.

Done.

• Line 381: aOR=8.5?

Correct (and changed accordingly). Thanks very much for detecting this mistake…

Reviewer #4:

 “The manuscript significantly improved after this revision. However, introduction and limitation sections unnecessarily long that need to be concise before final decision.”

Done (see reply to first comment of reviewer 3).

---

## [Decision Letter · Decision Letter 2]

19 Sep 2022

Over-indebtedness and health in Switzerland: a cross-sectional study comparing over-indebted individuals and the general population

PONE-D-21-26584R2

Dear Dr. Hämmig,

We’re pleased to inform you that your manuscript has been judged scientifically suitable for publication and will be formally accepted for publication once it meets all outstanding technical requirements.

Kind regards,

Sergio A. Useche, Ph.D.

Academic Editor

PLOS ONE

Reviewers' comments:

Reviewer's Responses to Questions

**Comments to the Author**

1. If the authors have adequately addressed your comments raised in a previous round of review and you feel that this manuscript is now acceptable for publication, you may indicate that here to bypass the “Comments to the Author” section, enter your conflict of interest statement in the “Confidential to Editor” section, and submit your "Accept" recommendation.

Reviewer #3: (No Response)

Reviewer #4: All comments have been addressed

2. Is the manuscript technically sound, and do the data support the conclusions?

Reviewer #3: Yes

Reviewer #4: Yes

3. Has the statistical analysis been performed appropriately and rigorously? 

Reviewer #3: Yes

Reviewer #4: Yes

4. Have the authors made all data underlying the findings in their manuscript fully available?

Reviewer #3: Yes

Reviewer #4: Yes

5. Is the manuscript presented in an intelligible fashion and written in standard English?

Reviewer #3: Yes

Reviewer #4: Yes

6. Review Comments to the Author

Reviewer #3: Well done! Only few minor suggestions:

* Line 320: 'most strongest". Delete "most".

* Line 273-276: too long sentence with resulting unclear message.

Reviewer #4: This manuscript addressed most of the comments that were raised during the review process. I recommend its publication.

7. PLOS authors have the option to publish the peer review history of their article (what does this mean?). If published, this will include your full peer review and any attached files.

Reviewer #3: **Yes: **Hans Bosma

Reviewer #4: No

---

## [Editor Report · Acceptance letter]

29 Sep 2022

PONE-D-21-26584R2 

Over-indebtedness and health in Switzerland: a cross-sectional study comparing over-indebted individuals and the general population 

Dear Dr. Hämmig:

I'm pleased to inform you that your manuscript has been deemed suitable for publication in PLOS ONE. Congratulations! Your manuscript is now with our production department. 

Kind regards, 

on behalf of

Dr. Sergio A. Useche 

Academic Editor

PLOS ONE